# Realizing nearly-free-electron like conduction band in a molecular film through mediating intermolecular van der Waals interactions

Xingxia Cui [1,6], Ding Han [1,6], Hongli Guo [1,6], Linwei Zhou [2], Jingsi Qiao [2], Qing Liu [1], Zhihao Cui [1], Yafei Li [1], Chungwei Lin[3], Limin Cao [1], Wei Ji [2], Hrvoje Petek [4] & Min Feng [1,5]

Collective molecular physical properties can be enhanced from their intrinsic characteristics by templating at material interfaces. Here we report how a black phosphorous (BP) substrate concatenates a nearly-free-electron (NFE) like conduction band of a $C_{60}$ monolayer. Scanning tunneling microscopy reveals the $C_{60}$ lowest unoccupied molecular orbital (LUMO) band is strongly delocalized in two-dimensions, which is unprecedented for a molecular semiconductor. Experiment and theory show van der Waals forces between $C_{60}$ and BP reduce the inter-$C_{60}$ distance and cause mutual orientation, thereby optimizing the $\pi$-$\pi$ wave function overlap and forming the NFE-like band. Electronic structure and carrier mobility calculations predict that the NFE band of $C_{60}$ acquires an effective mass of 0.53–0.70 $m_e$ ($m_e$ is the mass of free electrons), and has carrier mobility of ~200 to 440 $cm^2V^{-1}s^{-1}$. The substrate-mediated intermolecular van der Waals interactions provide a route to enhance charge delocalization in fullerenes and other organic semiconductors.

[1] School of Physics and Technology and Key Laboratory of Artificial Micro- and Nano-Structures of Ministry of Education, Wuhan University, Wuhan 430072, China. [2] Beijing Key Laboratory of Optoelectronic Functional Materials & Micro-Nano Devices, Department of Physics, Renmin University of China, Beijing 100872, China. [3] Mitsubishi Electric Research Laboratories, 201 Broadway, Cambridge, MA 02139, USA. [4] Department of Physics and Astronomy and Pittsburgh Quantum Institute, University of Pittsburgh, Pittsburgh, PA 15260, USA. [5] Institute for Advanced Studies, Wuhan University, Wuhan 430072, China. [6]These authors contributed equally: Xingxia Cui, Ding Han, Hongli Guo. Correspondence and requests for materials should be addressed to W.J. (email: wji@ruc.edu.cn) or to H.P. (email: petek@pitt.edu) or to M.F. (email: fengmin@whu.edu.cn)

The degree of electron delocalization in organic semiconductors is critical for their adoption in electronic and optoelectronic applications[1–3]. Electron transport, however, is usually facile through chemical bonds in conjugated organic materials, but is rarely optimal when the noncovalent intermolecular van der Waals (vdW) forces define the self-assembly and consequently, the intermolecular electronic coupling[4]. $C_{60}$ molecules, which have uncommonly large electron affinity and suitable electronic band gap, have been extensively investigated as zero- to three-dimensional (0-3D) organic semiconductors[5–9]. As a typical vdW material, $C_{60}$ molecules form solids through a balance of the Pauli type, electrostatic repulsive force and the attractive vdW force[10], which, however, does not enhance intermolecular electronic hybridizations. $C_{60}$ solids are characterized by flat electronic bands with inconsequential dispersions and hopping transport. Functionalization of $C_{60}$ by synthesizing fullerene derivatives has been investigated as a means to improve its electron transport, but the improvement was limited as the dispersions of the electronic bands were not effectively modified[11–14]. A radical improvement of electron transport, however, might be achieved if the assembly of $C_{60}$ molecules could increase the intermolecular electronic hybridizations.

Even though the vdW forces are weak compared to other bonding mechanisms, it can have a profound impact on the electronic properties. For example, a recent study has demonstrated that by controlling the vdW interactions in a bilayer graphene, it is possible to transform a nominally semiconducting material into a superconductor[15]. Our method to mediate the vdW interactions between $C_{60}$ molecules is through a substrate control of intermolecular interactions. It has been demonstrated that epitaxial growth of CO molecules on metals can impose an in-plane molecular compression that affects the intermolecular electronic interactions[16,17]. On metal surfaces, however, the high electron affinity of $C_{60}$ enables it to naturally draw electrons from its substrates and thereby to become metallic[18,19]. Thus, to retain and enhance its semiconducting properties, it is far more desirable to impose intermolecular interactions on $C_{60}$ molecules with an electronically inert substrate. From this point of view, black phosphorous (BP)[20,21], composed of atomic sheets that are held by vdW forces with alternating projecting "ridge" and subsiding "notch" atoms in a single layer, provides a unique adsorption platform for $C_{60}$ molecules with yet unknown consequences.

Here we show by scanning tunneling microscopy (STM) and theory that the BP substrate organizes $C_{60}$ molecules into a compressed monolayer and imposes a favorable orientation that optimizes the intermolecular π-π couplings, resulting in a nearly-free-electron (NFE) like lowest unoccupied molecular orbital (LUMO) band in $C_{60}$ monolayers. Such NFE-like LUMO band would dominate charge transport in $C_{60}$ assemblies. An NFE band in $C_{60}$ nanostructures has been attested through discovery of the superatom molecular orbitals (SAMOs)[22], and their strong hybridization. But SAMOs and their NFE bands are too high in energy to participate in charge transport unless $C_{60}$ molecules are endohedrally doped with metallic atoms[23]. NFE dispersion has been attributed to π orbitals of a molecular monolayer on metallic substrates, but its origin has later been reassigned to quantum confinement of a metal surface state, rather than the electronic hybridization among organic molecules[24–28]. The NFE conduction formed by π-π interactions is worthy of further exploration in $C_{60}$ or other organic molecules on inert substrates, because it realizes a long sought mode for charge transport for high-performance organic electronics and optoelectronics[4].

## Results

**Topography of $C_{60}$ monolayer on BP.** Figure 1a shows a topographic reconnaissance STM image of a $C_{60}$ monolayer island on BP, where $C_{60}$ molecules aggregate into a compact, highly ordered film. The top inset in Fig. 1a shows a topographic STM image, with atomic resolution, of the BP substrate where only the topmost ridge atoms are resolved while the lower notch atoms remain undetected within the dark waving lines. A rectangular surface unit cell is marked in the image with lattice constants $l_1$ and $l_2$ along the zig-zag and arm-chair directions, respectively. By analyzing the central position of $C_{60}$ molecules at the island edge (bottom right inset in Fig. 1a), we establish that they assemble above notches of the BP layer (noted by white dashed lines), while the exact adsorption sites will be elaborated later. Figure 1b presents a close-up STM topographic image of unoccupied states ($V_{bias} = +1.05$ V) of the $C_{60}$ monolayer, where each molecule appears as two bright lobes. As established in previous studies[18,29,30], the characteristic two-lobe structure corresponds to the unoccupied π-orbitals of two pentagons that are joined by a C = C bond (see Fig. 1c). Here, we mark the C = C bond with "h:h" (The "h:h" designation represents C = C bonds shared by the two side-by-side hexagons)[31]. This two-lobe intramolecular feature establishes that $C_{60}$ molecules sit on the surface in a highly ordered structure, where its two side-by-side hexagons and h:h joined pentagons point up (Fig. 1c).

In Fig. 1b, it is clear that along the zig-zag direction of the substrate, the h:h bonds are oriented in the same direction forming an equivalently oriented $C_{60}$ chain over the BP notches. In the adjacent $C_{60}$ chain, the h:h bonds are again oriented parallel to each other, but the h:h-h:h angle (indicated by φ in Fig. 1d) with respect to the original chain is 99.2 ± 0.4°, i.e., the h:h bonds orientation alternates between $C_{60}$ molecules in the arm-chair direction. The alternation of the h:h bonds are highlighted in Fig. 1d, where ball-stick models of $C_{60}$ are superimposed on the topography image with the h:h bonds marked in red. With this molecular arrangement, $C_{60}$ molecules form a centered supercell, which contains two molecules of different h:h orientations with lattice constants $a = 9.9 \pm 0.2$ Å (zig-zag) and $b = 17.2 \pm 0.2$ Å (arm-chair), corresponding to three and four times the substrate lattice constants along the zig-zag and arm-chair directions, respectively. The obtained intermolecular distances $d_1$ is nearly identical to $d_2$ with values of $9.9 \pm 0.2$ Å, and $d_3$ is of $10.0 \pm 0.2$ Å (Fig. 1d). These distance values imply a tighter packing of $C_{60}$ molecules in the monolayer than that in molecular crystals where the vdW distance is 10.04 Å. The distance is also smaller than commonly found in $C_{60}$ monolayer on most metal surfaces[29,32–36]. The STM image shown in the inset of Fig. 1b identifies six positions with weak, bright contrast around each $C_{60}$ molecule, as marked by arrows; two arrows belong to interfaces between $C_{60}$ molecules along the zig-zag direction (blue) and the rest are along the arm-chair direction (black). We will elucidate later how this interface contrast relates to strong intermolecular interactions unique to the $C_{60}$ monolayer on BP surface.

Figure 1e presents the most stable adsorption structure of an isolated $C_{60}$ molecule on a BP tri-layer based on DFT structural optimization. Consistent with the experiment, the bottom h:h bond (depicted in deep blue) lays over the BP notches such that each C atom interacts with two of the closest P atoms (dark pink). The relative orientation of the bottom h:h to the interacting P-P bond direction has an angle of $\theta = 11.0°$ (Fig. 1e). To get more insight into $C_{60}$-BP interactions, we calculate the differential charge density [DCD, total – (BP + $C_{60}$)] between $C_{60}$ and BP. The largest DCD of 0.0029 eÅ$^{-3}$ is about 0.001 of the largest pre-adsorption density of 2.1925 eÅ$^{-3}$, implying there is negligible

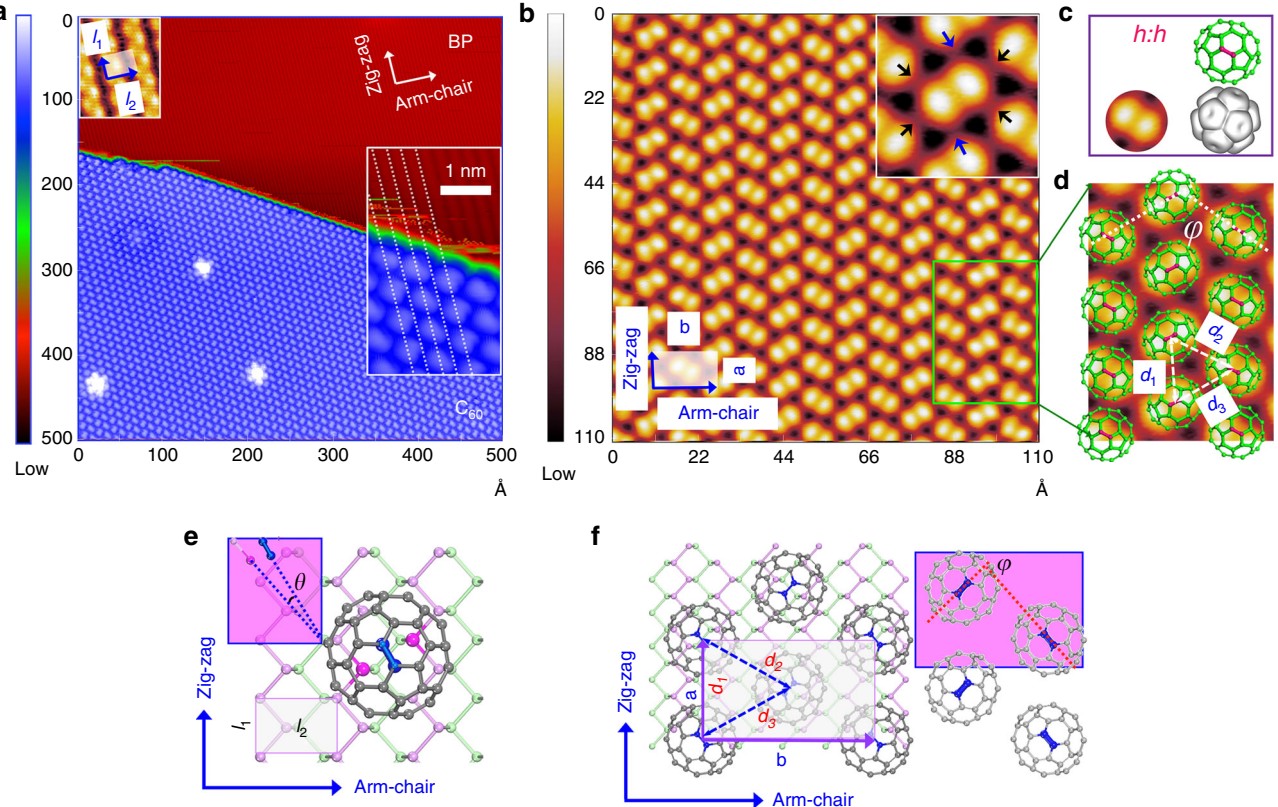

**Fig. 1** The adsorption structure and site of $C_{60}$ on BP. **a** STM image ($I = 30$ pA, $V_b = -1.55$ V) of an ordered $C_{60}$ monolayer island on BP. The upper left inset shows an atomic resolution STM image ($I = 60$ pA, $V_b = -0.60$ V) of the bare BP surface. Only the upper atoms of the topmost puckered BP layer are observed. The surface unit cell is marked with $l_1$ and $l_2$ representing the lattice constants along zig-zag and arm-chair directions, respectively. The lower right inset shows an enlargement at a termination of a $C_{60}$ island: The dashed white lines crossing the center of close-packed $C_{60}$ molecules extend to the location of "notches" of the BP surface. **b** A close-up STM image ($I = 30$ pA, $V_b = 1.05$ V) of $C_{60}$ on BP that shows two-lobes of intramolecular contrast. The inset images show six locations of moderately enhanced contrast around each $C_{60}$ molecule which can be categorized into two kinds by symmetry along the arm-chair direction (indicated by black arrows) and the zig-zag direction (indicated by blue arrows). **c** The model structure shows that $C_{60}$ adsorbs on the BP surface with the *h:h* bond of $C_{60}$ icosahedra pointing up. **d** Enlargement of the white rectangle in (**b**). $C_{60}$ ball-and-stick model is superimposed over $C_{60}$ images to highlight the experimental orientation of *h:h* bonds, which alternates with a relative angle of $\varphi = 99.2° \pm 0.4°$. **e** A ball-and-stick model of the most stable adsorption structure of isolated $C_{60}$ on BP obtained from DFT calculations. The rectangle represents the calculated unit cell of BP with $l_1$ and $l_2$ dimensions as in (**a**). The C and P atoms involved in the interaction are highlighted by deep blue and dark pink, respectively. The relative orientation of the of *h:h* bond to the BP substrate is indicated by $\theta$. **f** A ball-and-stick model of the most stable adsorption structure of $C_{60}$ lattice on BP obtained from DFT calculations. The unit cell of the $C_{60}$ lattice is: $a = 3\ l_1$; $b = 4\ l_2$ and the calculated relative orientation of *h:h* bonds is $\varphi = 97.3°$. The calculated intermolecular distances are $d_1 = d_3 = 9.9$ Å and $d_2 = 10.07$ Å. The STM images are obtained at 77 K

charge transfer between BP and $C_{60}$ and the interaction mainly occurs through vdW forces. The adsorption gains $-1.24$ eV energy per $C_{60}$ molecule and the C–P distances are 3.42~3.46 Å, consistent with the vdW force mediated adsorption energy of $C_{60}$ on graphene[37].

Figure 1f shows the most stable molecular configuration of the $C_{60}$ monolayer predicted by DFT. The orientation of $C_{60}$ molecules within the monolayer is similar to that of isolated $C_{60}$ molecules on BP, with the bottom *h:h* bond almost parallel to the substrate P-P bond with an angle of 1.6° between them. These results indicate that condensing $C_{60}$ into a monolayer does not strongly alter the local adsorption structure with respect to that of single $C_{60}$ molecules, implying that $C_{60}$-BP interactions dominate the monolayer formation. This assertion can also be quantified from an energetic perspective. The obtained absorption energy of $-1.36$ eV per $C_{60}$ molecule in the monolayer on BP is only 0.12 eV larger than that for an isolated $C_{60}$. The adsorption energy difference can be regarded as the energy gain due to inter-$C_{60}$ interactions, which are much weaker than that for the BP–$C_{60}$ interaction. Our calculations also reproduce the observed molecular arrangement with identical *h:h* bonds orientation

along the zig-zag direction and alternating *h:h* bonds along the arm-chair direction. The theoretical lattice constants $a$ and $b$ of 9.90 Å ($d_1$) and 17.20 Å and $\varphi$ of 97.3° reaffirm the compressed $C_{60}$ structure. We calculate that such in-plane lattice compression corresponds to an effective pressure of 1.8–2.2 GPa on a $C_{60}$ monolayer by the BP–$C_{60}$ interaction.

**Delocalized LUMO state of $C_{60}$ on BP surface**. Next, we examine the electronic structure that emerges from the $C_{60}$ monolayer compression. Figure 2a shows an exceptional STM topographic image acquired at a bias of approximately +0.80 V, where the intramolecular contrast of $C_{60}$ vanishes to be replaced by a wave-like supramolecular contrast. The STM image has a herringbone stich knitted fabric pattern with electron density distribution extending deep between the molecules, indicating that the contrast at this energy is dominated by intermolecular electron delocalization. The vanishing of the intramolecular contrast in delocalized 0D, 1D, and 2D structures has been observed for SAMOs of $C_{60}$[22,23], but such images have not been reported for $\pi$-orbitals forming the electronic gap.

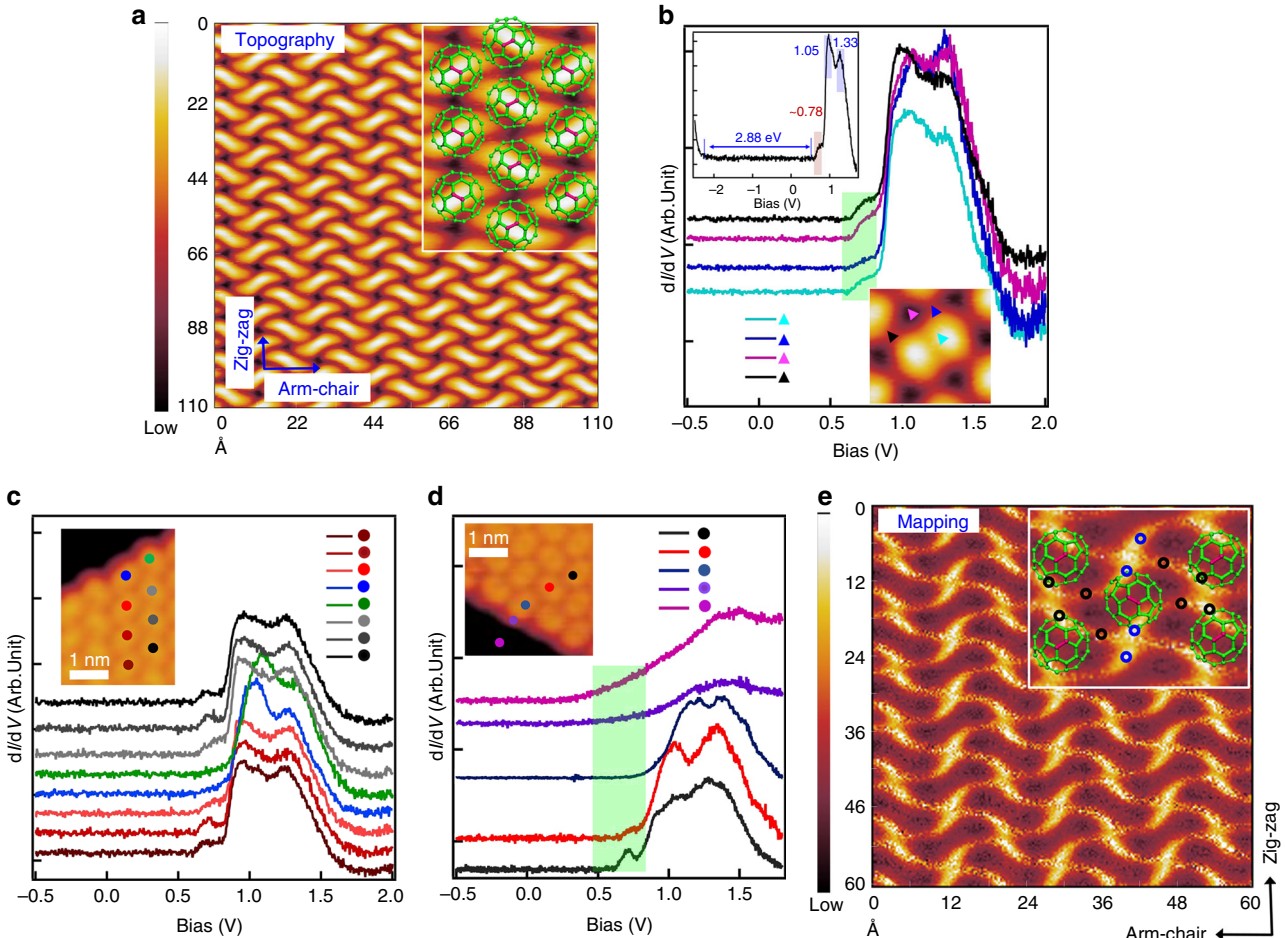

**Fig. 2** Highly delocalized LUMO state of $C_{60}$ on BP. **a** STM image of a $C_{60}$ monolayer island on BP obtained with $V_b = 0.80$ V and $I = 30$ pA. The image appears like a 2D herringbone stich pattern knitted fabric. $C_{60}$ ball-and-stick models are superimposed on the STM image in the inset to relate the molecular structure of $C_{60}$ with the wave-like contrast. **b** Position-dependent STS spectra obtained on a typical $C_{60}$ molecule within the island. The STS data are recorded at locations around a $C_{60}$ molecule shown in the STM image. The STS $dI/dV$ spectra in the inset are taken over a larger energy range, to record the band gap of $C_{60}$ molecules. The main unoccupied features due to the LUMO are highlighted in the inset. A low energy shoulder, which has not been observed from previous work, is highlighted by the light-green rectangle. **c** Position-dependent STS spectra obtained for $C_{60}$ molecules approaching an edge of a monolayer island. The colored dots correlate the STS spectra with molecule locations; spectra show that the shoulder feature gradually fades when approaching the island edge (blue and green spectra). **d** Position-dependent STS spectra acquired on the locations across the $C_{60}$/BP edge. The light-green rectangle highlights the energy region where the spectroscopic shoulder feature is observed. (**e**) STS spectroscopic image recorded monitoring the shoulder feature in (**b**) at 0.80 V. The DOS distribution appears delocalized in both the arm-chair and zig-zag directions skirting the indicated $C_{60}$ molecule centers. The black circles in the enlarged image of the insert indicate the four arms connecting $C_{60}$ molecules in the arm-chair direction. The blue circles indicate the two arms in the zig-zag direction. The arms of $C_{60}$ knit the 2D delocalized electronic net of the monolayer. The STM topography images are obtained at 77 K and the $dI/dV$ spectra and the $dI/dV$ mapping at 4.5 K

We establish the origin of the delocalized electronic structure by recording scanning tunneling spectroscopy (STS) spectra. The inset in Fig. 2b shows a spectrum measured between the valence and conduction bands. It indicates that the $C_{60}$ monolayer is semiconducting with a bandgap of 2.88 eV, very close to that of a $C_{60}$ bilayer or multilayer[38]. The band gap value and the location of the Fermi level within it confirms that the $C_{60}$ monolayer experiences negligible charge transfer upon adsorption on BP. Figure 2b provides STS spectra of the LUMO region, which are measured at four different positions on and around an individual $C_{60}$ molecule selected by considering the symmetry of the adsorption structure (Fig. 2b inset). The spectra are similar at these four sites and show a shoulder peak at the bottom of the LUMO resonance (from 0.67 to 0.82 eV), which distinguishes them from previous measurements. The two sharp peaks, residing at 1.05 and 1.33 eV, respectively, are typical in previously reported LUMO bands for metal supported $C_{60}$ bilayers[38]. The

less intense shoulder feature, to which we attribute the herringbone structure of the STM topographic image at 0.8 eV in Fig. 2a, is, however, exceptional to the $C_{60}$ monolayer on BP.

This newly observed shoulder feature in Fig. 2b originates from the electronic hybridization between π-orbitals of $C_{60}$ molecules, as established by a series of spectra measured over single $C_{60}$ molecules in Fig. 2c. The shoulder intensity gradually diminishes as the measuring location approaches the island edge and grows back when the tip returns to the island interior. STS spectra are also acquired on the locations across the $C_{60}$/BP island edge, as shown in Fig. 2d. It is evident that the spectroscopic shoulder feature of $C_{60}$ within the island terminates at the edge $C_{60}$ molecules; after leaving the $C_{60}$ island edge, the spectra show the typical bare BP character[21]. These spectra, thus, compellingly establish that the shoulder belongs to a $C_{60}$ bonding LUMO band, whose density is expected to be maximum within the island bulk and to decrease upon approaching the edge. This conclusion is

further confirmed by the spectroscopic d$I$/d$V$ map recorded at 0.80 eV (Fig. 2e), where there are appreciable electron densities around the edges of the herringbone topographic contrast. Specifically, the contrast involves four extended arms of $C_{60}$, two to the left and two to the right along the arm-chair direction (indicated by the black circles in the inset of Fig. 2e). These arms link four adjacent $C_{60}$ molecules in the arm-chair direction. In the zig-zag direction, there is also a high intensity contrast (indicated by the blue circles in the inset of Fig. 2e) that links two vertically adjacent $C_{60}$ molecules with one arm on each side. These six arms are responsible for the weak bright contrast observed around single $C_{60}$ molecules mentioned in Fig. 1b. Importantly, the electronic interaction corresponding to the arms form a delocalized 2D electron density network, in sharp contrast to the d$I$/d$V$ mapping images of the two commonly observed 1.05 and 1.33 eV LUMO states (Supplementary Fig. 1) where electron densities are localized on single molecules.

**Electronic hybridization among $C_{60}$ on BP surface.** We confirm the experimentally observed formation of the delocalized LUMO band by reproducing it with DFT calculations. The calculations are performed for both the $C_{60}$ monolayers with BP substrate and without BP substrate using the supercell shown in Fig. 1f. As shown in Supplementary Fig. 2, the energies and dispersions of the LUMO bands of two $C_{60}$ molecules in the supercell does not appreciably change upon the removal of BP substrate. Furthermore, removing the BP substrate does not cause appreciable change in the partial density of states of $C_{60}$ monolayer (Supplementary Fig. 3). In light of these comparisons, we conclude that DFT results of the isolated $C_{60}$ monolayer without BP substrate fully capture the essential features of the NFE band formation. As shown in Supplementary Fig. 2a, the inclusion of the BP substrate in the electronic structure calculation introduces several substrate bands in the proximity of the $C_{60}$ LUMO bands, which may blur the NFE LUMO band we intended to present. Therefore, we only provide DFT results of the fully relaxed $C_{60}$ monolayer without BP substrate in the following discussions.

Figure 3a plots the calculated density of states (DOS) of the unsupported $C_{60}$ monolayer. The LUMO region spans an energy range of over 0.5 eV, and consists of a shoulder and two dominant peaks at higher energy; this well reproduces the three features observed in our d$I$/d$V$ spectra (Fig. 2b). The band structure plot (Fig. 3b) establishes the band contributions to the DOS peaks. For an isolated $C_{60}$, the LUMO orbitals are triply-degenerate (Supplementary Fig. 4)[39]. LUMO-a, -b, -c derive from the LUMO orbitals of a free molecule when the degeneracy is broken by nonisotropic environment such as caused by the intermolecular and molecule-substrate interactions. The LUMO-a, -b, -c label is normally used to identify the three orbitals that can be identified when the perturbations are sufficiently strong, to distinguish their origin from lowest energy state to the higher ones[19]. Figure 3b is the calculated band structure of $C_{60}$ monolayer with the structures shown in Fig. 1f. Here, the supercell contains two $C_{60}$ molecules and those LUMO orbitals thus form six LUMO bands (three pairs of bonding–antibonding bands) in the monolayer. The dispersion and the energy levels of those bands are determined by $C_{60}$–$C_{60}$ interactions within the monolayer. The lowest unoccupied energy at the Γ point belongs to the LUMO-a bonding band, which results into the shoulder in the DOS plot (marked with the violent circle in Fig. 3a) and corresponds to the shoulder (~0.78 eV) recorded in our d$I$/d$V$ STS spectra (Fig. 2b). Those two higher-energy peaks marked with cyan and light orange circles in the DOS plot shown in Fig. 3a are ascribed to the LUMO-b, -c bonding bands and the LUMO-a, -b, -c antibonding bands, respectively, which reproduce

the two major d$I$/d$V$ peaks at 1.05 and 1.33 eV (Fig. 2b). More details on assignments of the two sharp peaks in the d$I$ / d$V$ STS spectra are presented in Supplementary Fig. 5.

The compelling result of the band structure calculation is that the LUMO-a bonding band is highly dispersive, characterized with the electron effective masses ($m^*$) of 0.70 $m_e$ and 0.53 $m_e$ along the Γ-$X$ (zig-zag) and Γ-$Y$ (arm-chair) directions, respectively (Table 1). We further confirm the rather small DFT calculated $m^*$ in $C_{60}$ monolayer by performing complementary tight binding calculations (Supplementary Note 1 and Supplementary Fig. 6). These calculations give a bandwidth of 0.64 eV for the LUMO-a band and $m^*$ values of 0.96 $m_e$ along the Γ-$X$ (zig-zag) and 0.32 $m_e$ along Γ-$Y$ (arm-chair) directions. These results indicate electrons excited or injected into the 2D delocalized LUMO-a band behave like free electrons, which is consistent with the 2D herringbone network appearance of the d$I$/d$V$ image (Fig. 2e). A real-space plot of the wave function norm of the 0.57 eV shoulder state (Fig. 3c) reveals wave function overlaps among the four arms of each $C_{60}$ molecule along the arm-chair direction and two arms along the zig-zag direction, accounting for the experimental observation (Fig. 2e). Figure 3d reveals that 2D delocalized charge distribution is a two-layer structure parallel to the surface in real space. The LUMO-b and LUMO-c bands have larger effective masses (Table 1) and their wave functions are more localized to the $C_{60}$ molecules.

**Origin of the NFE LUMO-a band.** A highly delocalized band, such as LUMO-a, is unprecedented for a molecular monolayer bound by vdW forces; it is also significant that it is specific to LUMO-a only. To pinpoint the origin of the NFE bands, in Fig. 3e we first examine the top and side views of probability densities of the LUMO-a, -b, -c orbitals of an isolated $C_{60}$ molecule, with the molecular orientation that is related to experiments. The difference between the LUMO orbitals can be seen by their probability densities with respect to the molecular plane. Specifically, the LUMO-a orbital is primed to hybridize within the molecular plane (Fig. 3e), but the LUMO-b and -c have densities projecting above and below the plane, and hence their intermolecular hybridizations within the molecular plane are weak (Fig. 3e). The calculations show that the unique $C_{60}$ lattice on BP surface favors the in-plane hybridization of probability density of the LUMO-a orbital resulting in the NFE band.

The detailed features of the in-plane hybridization can be recognized by plotting the intermolecular DCD of LUMO-a of the monolayer. Figure 4a shows how the delocalized band forms through intermolecular density sharing between $C_{60}$ molecules. In particular, the hybridization occurs at interfaces between molecules along the arm-chair and zig-zag directions (indicated by black and blue circles). This is consistent with the total of six extended arms of $C_{60}$ intermolecular interactions that are observed in d$I$/d$V$ mapping image (Fig. 2e). The chirality of the arms along the zig-zag direction is also well reproduced.

**Dependence of the NFE LUMO-a band on $C_{60}$ monolayer topography.** The single molecule wave functions and their interactions give clues for the formation of the NFE LUMO-a band. To illustrate the roles of the unique compressive $C_{60}$ molecular lattice that is imposed by the BP, i.e., the intermolecular distance and orientation within the 2D lattice, we calculate the DOS of a $C_{60}$ monolayer, without the substrate, for different lattice constants, $a$ and $b$, and mutual orientation angles, $φ$, that are indicated in Fig. 1f. The calculation results are presented in Fig. 4b–d. Figure 4b, c shows that the lattice constant compression along the arm-chair direction, $b$, is critical for enhancing the DOS of the NFE-shoulder state. The NFE-shoulder

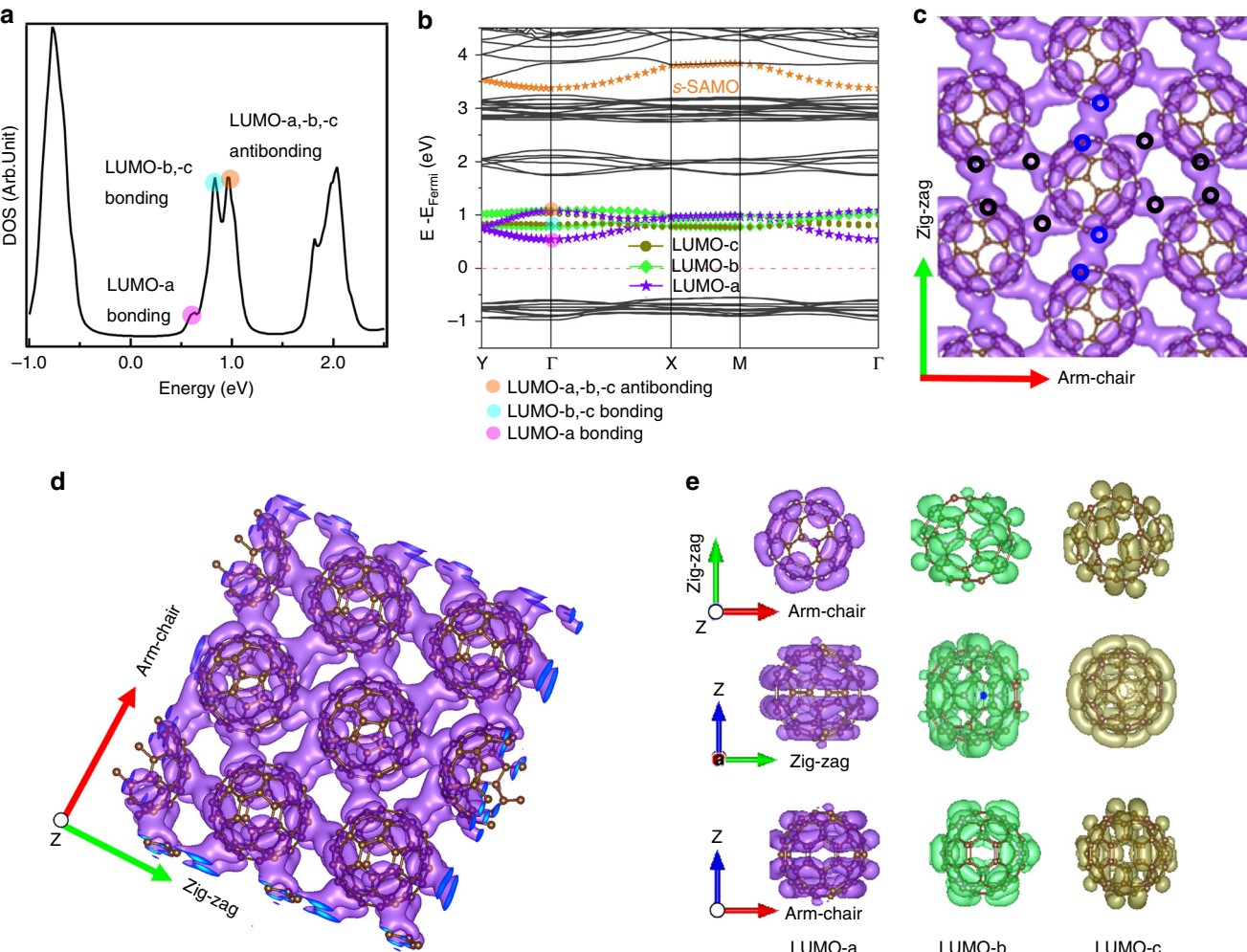

**Fig. 3** Nearly-free-electron like LUMO band of $C_{60}$ monolayer on BP. **a** DFT calculated DOS of an unsupported $C_{60}$ monolayer lattice with the same geometry as on a BP support, shown in Fig. 1f. The calculated DOS reproduces the two major peaks and one shoulder for the LUMO complex in the STS data. **b** The calculated band structure of the $C_{60}$ lattice in Fig. 1f. The three pairs of bonding–antibonding bands of LUMO-a, -b, and -c are represented by purple, bright green and dark green colors, respectively. The band formed by the superatom s-SAMO orbitals, is indicated in orange. Most importantly, the LUMO-a band is as dispersive as the NFE s-SAMO band. **c** The spatial distribution of wave function square of the LUMO-a bonding band at the Γ point. The distribution shows a 2D delocalized character as observed in the STS image (Fig. 2e). The black circles highlight the four arms between adjacent molecules along the arm-chair direction, where the bonding interaction dominates. The blue circles highlight the two arms along zig-zag direction. The iso-surface value of the image is 0.0005 and the largest value is 0.02. **d** 3D image of Fig. 3c shows that the 2D delocalized probability density distribution has two layers in real space. **e** The top and side views of spatial distributions of LUMO-a, -b, -c orbitals of an isolated $C_{60}$ molecule with the same geometry as on BP surface. Z represents the direction vertical to the plane determined by the arm-chair and zig-zag directions. It is evident that the in-plane C atoms in the LUMO-a state have high density in the molecular plane and thus contribute to intermolecular interactions. By contrast The LUMO-b and -c have density that projects above and below the plane, and hence, their intermolecular hybridization is weak

## Table 1 Effective masses and mobilities

| | Effective mass, $m^*$ (zig-zag) | Effective mass, $m^*$ (arm-chair) | Mobility ($cm^2 V^{-1} s^{-1}$) (zig-zag) (band-like transport model) | Mobility ($cm^2 V^{-1} s^{-1}$) (arm-chair) (band-like transport model) | Mobility ($cm^2 V^{-1} s^{-1}$) (arm-chair) (hopping transport model) |
|---|---|---|---|---|---|
| LUMO-a | 0.70 | 0.53 | 201–224 | 339–440 | 14.7 |
| LUMO-b | 1.37 | 3.18 | 152–177 | 160–248 | 8.3 |
| LUMO-c | 3.16 | 3.82 | 135–229 | 173–281 | 4.4 |

The effective masses and mobilities (band-like transport and hopping transport mechanisms) of electron along different transport directions (zig-zag and arm-chair) for the LUMO-a, -b, -c bands

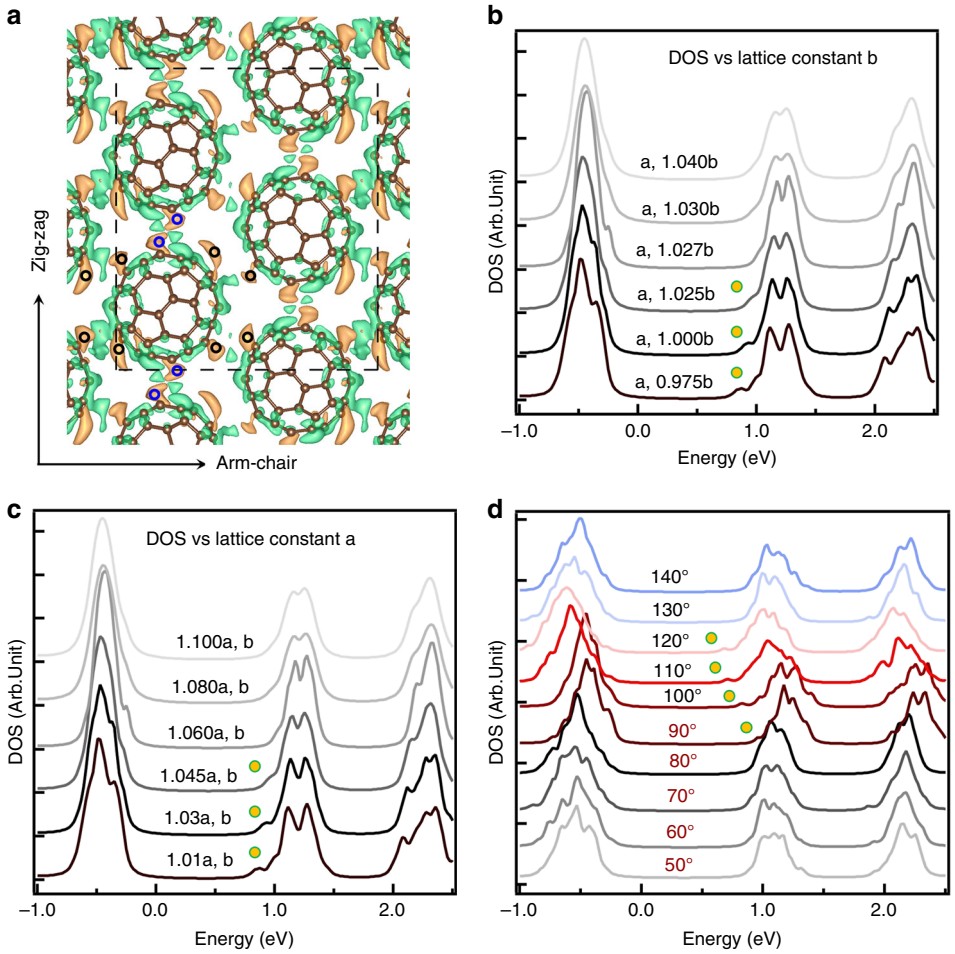

**Fig. 4** The origin of the nearly-free-electron like LUMO-a band in a $C_{60}$ monolayer. **a** The intermolecular differential probability density [(total − ($C_{601}$ + $C_{602}$ + $C_{603}$ + $C_{604}$); $C_{601}$, $C_{602}$, $C_{603}$, $C_{604}$ represent the four $C_{60}$ molecules in the calculated unit cell indicated by the black dashed rectangle] of a $C_{60}$ monolayer in Fig. 1f. The green and orange iso-surfaces represent the electron donation and accumulation regions, respectively. The results show a probability density sharing redistribution where one arm donates charge and the neighboring molecule arm accepts it. It can be seen that the intermolecular hybridization occurs between molecules that are arranged such that their relative h:h bond orientations alternate. The iso-surface value of the image is 0.0001 and the maximum value is 0.033. **b**, **c** The appearance of the NFE LUMO-a bonding band shoulder by compression of the lattice constant, *a*, along the zig-zag and *b*, the arm-chair direction; the h:h bond orientation $\varphi$ is kept constant. The yellow dots with green circle indicate the NFE band density. The trend is that the NFE-shoulder disappears as the lattice constant increases with distortion in *b* being more sensitive than *a*. **d** The appearance of the NFE LUMO-a bonding band shoulder by varying the h:h relative orientation angle $\varphi$ with the lattice constants *a* and *b* kept constant. The yellow dots indicate the NFE state exists for only a small range of $\varphi$ from 90° to 120° overlapping with the experimentally observed $\varphi = 99.2 \pm 0.4°$. The results show that both the relative orientation of $C_{60}$ molecules and the intermolecular distance play important roles in the formation of the NFE state

gradually fades out upon increasing *b* and disappears for $1.03 \times b$, and larger, corresponding to $d_2$ of ~10.11 Å. The shoulder state is less sensitive upon elongating the lattice constant, *a*, because in the zig-zag direction the lattice is more heavily compressed than the arm-chair direction when the $C_{60}$ monolayer lattice is constructed by the BP template.

Moreover, the calculations show that the lattice compression alone is not sufficient for producing the NFE band; the intermolecular orientation matters. Figure 4d shows the DOS of the monolayer upon varying the intermolecular orientation angle $\varphi$, while keeping the lattice at a constant *a* and *b*. Varying $\varphi$ has a more complex effect on the distribution of the DOS amplitudes than for varying lattice constants. It is evident, however, that when $\varphi$ is in the range from 90° to 120°, the LUMO peak distribution becomes much broader and small peaks clearly appear in the energy region below the noninteracting LUMO states. The NFE band features are only pronounced within a certain range of angles where the LUMO-a orbitals of adjacent $C_{60}$ molecules have a stronger π-π wave function overlaps

(Supplementary Fig. 7). For the $C_{60}$ monolayer on BP, the theoretically and experimentally observed $\varphi$ are 97.3° and 99.2°, respectively. These angles are within the angular window of an appreciable intermolecular interaction, indicating that BP substrate imposes a template that favors inter-$C_{60}$ hybridization. This angle dependence also explains why within $C_{60}$ bulk solids where the intermolecular distance of 10.04 Å is close to the critical distance[34], the dispersive LUMO bands have not been found. In the case of $C_{60}$ solids, molecules rotate at room temperature, resulting in a random orientation of adjacent molecules that does not support band formation and delocalization. The results explain how the relative $C_{60}$ orientation and compression that are enforced by the BP substrate lead to the intermolecular electronic hybridization of the LUMO-a into the NFE band that is imaged in experiments.

**Carrier mobility**. Next, we examine possible novel physical properties that are embodied in the dispersive band of the $C_{60}$ monolayer. The discovery and imaging of the delocalized

LUMO-a band suggest that it could enable band-like transport, rather than the well documented hopping mode found in $C_{60}$ aggregates[40]. Table 1 shows the effective masses and calculated electron mobilities using phonon-limited band-like and carrier hopping transport models[40,41]. The actual carrier mobility is difficult to measure by conventional means because the conduction band minimum of BP is below that of $C_{60}$. The theoretical mobilities of LUMO-a are 201 to 224 $cm^2\,V^{-1}\,s^{-1}$ at 300 K along zig-zag and increase up to 339 to 440 $cm^2 V^{-1} s^{-1}$ in the arm-chair direction. These values are comparable to some covalent 2D inorganic solids such as $MoS_2$[42], which is surprisingly high for an organic molecular monolayer that forms through vdW interactions. Carrier mobilities are affected by various processes, thus the predicted values can be different than the measured ones. Nevertheless, it is meaningful to compare theoretical values for a range of molecular layers. Both the calculated hopping (14.7 $cm^2\,V^{-1}\,s^{-1}$) and the band-like mobility (201–223 $cm^2\,V^{-1}\,s^{-1}$) of LUMO-a are several times the values calculated for pentacene mono- (hopping, ~10–15 $cm^2\,V^{-1}\,s^{-1}$) and bi-layers (band-like, 9–65 $cm^2\,V^{-1}\,s^{-1}$) transport[40], where the molecular arrangement does not favor effective π-π interaction in the conduction band. The band-like mobilities of LUMO-b and -c also appear to be high. However, the band-like transport mechanism is probably not applicable owing to their modest hybridization within the monolayer. Their hopping mobilities are calculated to be 4.4–8.3 $cm^2\,V^{-1}\,s^{-1}$, consistent with previous experimental and theoretical values for $C_{60}$ bands with unfavorable intermolecular interactions[9].

## Discussion

We have provided the experimental evidence and the theoretical understanding for an unusually dispersive NFE-like conduction band of a $C_{60}$ monolayer, which we attribute to a favorable vdW templating of $C_{60}$ molecules that enhance π–π interactions. In the literature on organic electronics, there are many discussions of π-π interactions, and how they might enhance charge transport, without much actual evidence that NFE band could be induced by π-π interactions. In this work, we show that it does happen for a molecule like $C_{60}$ if the π-π interactions are favored by an appropriate substrate template. The first realization of the NFE conduction band in a $C_{60}$ monolayer presents a new target for exploring the influence of structure on strongly correlated phenomena, e.g., insulating, metallic, superconducting and even magnetic phases[5,6,43,44], in this material. The fact that the π-π vdW interactions could trigger the NFE-like electronic band formation implies that templating of intermolecular interactions by vdW forces on otherwise weakly interacting substrates[45] provides a promising strategy for tailoring remarkable electronic properties in organic materials for electronics and optoelectronics[2–4,46].

## Methods

**Sample preparation and STM/STS measurements.** The BP crystals are self-grown using a chemical vapor transport (CVT) method [red phosphorus, tin iodide ($SnI_4$), and tin powders as the starting materials] in a two-zone tube furnace in a temperature gradient of 600–540 °C. The STM and spectroscopy experiments are carried out in an ultrahigh vacuum low temperature STM system (CreaTec). Prior to STM experiments, the BP crystals are cleaved in-situ in a preparation chamber under ultrahigh vacuum at room temperature (RT). $C_{60}$ molecules (99.9% purity, Aldrich) are sublimated from a resistively heated evaporator onto a freshly prepared BP surface. The room-temperature sample is then immediately transferred into the STM chamber, and cooled down to 77 and/or 4.5 K. The adsorption structure of $C_{60}$ molecules within the monolayer is stable at $LN_2$ temperature. STM topographic images are acquired in the constant-current mode. The d$I$/d$V$ spectra are measured using the standard lock-in technique with a bias modulation of 15 mV at 321.333 Hz. The STM tips are chemically etched tungsten or mechanically cut Pt-Ir wires, which are further calibrated spectroscopically against the

Shockley surface states of cleaned Cu(111) or Au(111) surfaces before being utilized on $C_{60}$/BP.

**DFT calculations.** First-principles DFT calculations are performed using the Vienna Ab initio Simulation Package[47,48]. The Perdew-Burke-Ernzerhof (PBE) exchange-correlation functional[49] along with the projector-augmented wave potentials[50,51] are used for the self-consistent total energy calculations and geometry optimization. The energy cutoff for the plane-wave basis is set to 400 eV for all calculations. In the band structure calculation, a $7 \times 5 \times 1$ K-mesh is adopted to sample the first BZ of the conventional unit cell of the slab of $C_{60}$; 60 points are collected along each high symmetry line in reciprocal space. vdW interactions are considered at the vdW-DF level when optimizing the system geometry[52,53]. With the optB86b-vdW exchange functional, the optimized lattice constant is in good agreement with the experimental values for BP [3.3 Å (zig-zag) and 4.3 Å (arm-chair)]. The unit cell is optimized fully by letting all atoms in the cell to relax until the residual force per atom is <0.01 eV Å$^{-1}$. A 15 Å vacuum layer separates the neighboring slabs of $C_{60}$ monolayer in vacuum and when deposited on BP. We use a slab model composed of 3 layers of BP to simulate the BP surface.

**Charge carrier mobility calculations.** Phonon-limited carrier mobility in $C_{60}$ monolayer with a finite effective thickness $W_{eff}$ is expressed as Eq. 1:[41,54–57]

$$\mu_{film} = \frac{\pi e \hbar^4 C_{film}}{\sqrt{2}(k_B T)^{3/2}(m^*)^{5/2}(E_1^i)^2} F. \tag{1}$$

Here, $m^*$ represents the effective mass along the transport direction and $E_1$ is the deformation potential constant of the VBM (hole) or CBM (electron) along the transport direction; it is determined by $E_1^i = \Delta V_i/(\Delta l/l_0)$. Here $\Delta V_i$ is the energy change of the $i$th band under proper compressive and tensile strain (by a step $\frac{\Delta l}{l_0} = 0.5\%$), $l_0$ is the corresponding lattice constant along the transport direction, and $\Delta l$ is the deformation of the lattice constant. The variable $C_{film}$ is the elastic modulus of the longitudinal strain in the propagation direction, which is derived by $(E - E_0)/V_0 = C(\Delta l/l_0)^2/2$; $E$ represents the total energy and $V_0$ represents the lattice volume at the equilibrium for a 2D system. A crossover function $F$ bridges the 2D and 3D cases, and is estimated by Eq 2:

$$F \equiv \frac{\sum_n \left\{ \frac{\sqrt{\pi}}{2}[1 - \text{erf}(\Omega(n))] + \Omega(n)e^{-\Omega^2(n)} \right\}}{\sum_n [1 + \Omega^2(n)]e^{-\Omega^2(n)}}, \tag{2}$$

where

$$\Omega(n) \equiv \sqrt{\frac{n^2\pi^2\hbar^2}{2m^* W_{eff}^2 k_B T}}, \tag{3}$$

The $\Omega(n)$ (Eq. 3) represents an error function and the summation over integer $n$ is due to quantum confinement along the z-direction. Effective thickness of the film, $W_{eff}$, is expressed by Eq. 4:

$$\frac{1}{W_{eff}} = \int_{-\infty}^{+\infty} P_i(z)P_f(z)dz = \sum_n \frac{\rho_i^n(z)}{N\Delta z} \cdot \frac{\rho_i^n(z)}{N\Delta z}\Delta z, \tag{4}$$

Here, $P(x)$ is the electron probability density along the z direction. We divided the space along the z direction into $n$ parts by $\Delta z$. Variable $\rho^n(z)$ is the sum of the number of electrons $n$th region along the z direction. Here, $N$ is the total number of valence electrons in the film, and $i$ and $f$ represent equilibrium and deformed films, respectively. All lattice structural properties and electronic structures in the calculation of carrier mobilities are obtained using the optB86b-vdW functional. The temperature used for the mobility calculations is 300 K.

To evaluate the mobility due to the electron/hole hopping between the adjacent molecules, we apply a treatment expounded by Deng and Goddard[58], which is based on the Marcus-Hush theory[59,60]. The charge transfer rate, $W_{ij}$, between the $i$-th and $j$-th pair of molecules is evaluated as Eq. 5:

$$W_{ij} = \frac{V_{ij}^2}{\hbar} \left( \frac{\pi}{\lambda k_B T} \right)^{1/2} \exp^{-\frac{\lambda}{4k_B T}}, \tag{5}$$

where $V_{ij}$ is the coupling matrix element and $\lambda$ is the reorganization energy. By assuming that the charge transfers between pairs of molecules are independent of each other, we can obtain the diffusion constant as Eq. 6:

$$D = \left\langle \frac{1}{2d} \sum_j r_{ij}^2 W_{ij} P_{ij} \right\rangle_{i=1,2}; \quad P_{ij} = \frac{W_{ij}}{\sum W_{ij}}, \tag{6}$$

where for $\langle...\rangle_{i=1,2}$ we take the mean value over two independent molecules in a unit cell, and $j$ runs over six neighboring molecules to the i-th molecule. $r_{ij}$ is the distance between adjacent molecules and $d = 2$ is the system dimension. The Einstein's relation gives the mobility in the form of Eq. 7:

$$\mu = \frac{e}{k_B T} D. \tag{7}$$

We take the reorganization energy of single $C_{60}$ molecule to be $\lambda = 0.06$eV[61,62].

## Data Availability
The data that support the findings of this study are available from the corresponding authors upon reasonable request.

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

## Acknowledgements

This work is financially supported by the National Key R&D Program of China (Grant Nos. 2017YFA0303500, 2017YFA0303504); the Strategic Priority Research Program of Chinese Academy of Science (Grant No. XDB30000000); the National Natural Science Foundation of China (Grant Nos. 11574364, 11622437 and 61674171), the Fundamental Research Funds for the Central Universities of China and the Research Funds of Renmin University of China (Grant No. 16XNLQ01). Calculations are performed at the Supercomputer Center of Wuhan University and Environmental Molecular Sciences Laboratory at the PNNL, and Physics Lab of High-Performance Computing of Renmin University of China. The contributions from M.F. and H.P. thank partial support from DOE-BES Division of Chemical Sciences, Geosciences, and Biosciences Grant No. DE-SC0002313, and Petek thanks the Luo Jia Visiting Chair Professorship.

## Author contributions

L.C. synthesized BP crystals. X.C., D.H., Q.L., Z.C. and Y.L. performed STM experiments. H.G., L.Z. and J.Q. performed theoretical calculations. M.F. conceived and provided advice on the experiment and analysis. W. J. conceived and provided advice on theoretical calculation. M.F., C.L., W.J. and H.P. participated in the data discussion. M.F., W.J. and H.P. wrote the paper with contributions from all the authors.

## Additional information

**Competing interests:** The authors declare no competing interests.

