## [Peer Review File · Nature Communications]

Reviewers' comments:

Reviewer #1 (Remarks to the Author):

In this paper, Cui et al. perform joint STM and DFT study on C60 molecular crystals epitaxially grown on 2D surface BP. They observe a delocalized molecular state located below the single-molecule state due to the closer molecular packing imposed by the compressing of BP substrate. Further, they are able to image delocalized molecular orbital, which is an unconventional feature for molecular semiconductors. DFT calculations are able to reproduce the molecular structure and capture the main experimental findings. In particular, DFT shows that the band dispersion is more than 0.5eV, with surprisingly low effective mass and high mobility. This is a nice example of how molecule-2D interaction strongly modifies the native properties of molecular crystals in a good way. The paper is well structured and clearly written, and experimental evidence is solid. I'm in favor of its publication in Nature Communications, however the following important issues have to be addressed properly.

1. In tight binding calculations, the transfer integral (t) between nearest neighbor plays important roles in determining the band widths and effective mass. To first order approximation, it should scale with the splitting between the single molecule orbital and collective band from interaction (named NFE in the paper), which is $\sim 0.2\text{eV}$ from STS. What is the calculated value of t ? The effective mass of $0.53m$ is remarkably small, compared to many inorganic semiconductors like MoS2 ($0.45m$). However, in those inorganic semiconductors, the transfer is one order of magnitude larger than C60 here. Therefore the low effective mass is not well justified. More in-depth discussion about transfer integral, band dispersion and effective mass is needed.
2. Again regarding t , the free standing C60 crystals have inter-molecular distance of 10.04 \AA , while the in current work it is reduced to $10.00 \pm 0.2 \text{ \AA}$. What's the effect of this 4% on t ?
3. What is LUMO a/b/c, are these the lowest LUMOs or just three random orbitals selected by the authors? Why select these three orbitals, and which orbitals do the two prominent peaks near 1.1eV correspond to?
4. I notice that the "arms" in experiments have different angles from theoretical calculations (see the difference between Fig. 2d inset and Fig. 3c), what is the origin of such inconsistency?
5. In calculating the delocalization of orbitals the substrate BP was not included, and therefore the substrate-molecule interaction was neglected. However, according to the calculations this interaction is much stronger than inter-molecular interactions, especially for LUMO-b and c according to fig. 3e. The authors claim that "The band structure calculations of the C60 monolayer are performed without the BP substrate because they are nearly identical in the band gap region", however Fig. S2 does not show band structure calculations with BP substrate.
6. Minor point: page 6 last line, green should have been blue.

Reviewer #2 (Remarks to the Author):

Cui et al report remarkable observations about the electronic structure of a C60 monolayer on a black phosphorous substrate. They find supramolecular patterns in topography and conductance that indicate delocalization of electrons in the unoccupied density of states in this adsorption system. DFT calculations suggest that these patterns arise due to very strong intermolecular pi-pi overlap that gives a free-electron like 2D band. Evidently, the black phosphorous substrate compresses the fullerene monolayer so that electronic coupling is enhanced beyond what is typically found in fullerene films.

The experimental observations are stunning enough that the paper should probably appear in a high profile journal like Nature Communications. Prior to publication, I recommend that the authors include STS spectra measured on the black phosphorous substrate. It is a glaring omission to not see this

data, particularly in Figure 2c where the position-dependent spectroscopy stops at the edge of the C60 layer. It should be extended to the BP region beyond for direct comparison.

Obviously this data would strengthen the assertion that the substrate is electronically inert and merely structurally templates the high degree of compression in the monolayer. It needs to be explicitly shown that there are no relevant substrate surface states.

On a related note, earlier reports of NFE behavior in molecular monolayers (Temirov et al Nature 444, 350 (2006)) were erroneously attributed to molecules alone and were only later attributed to a shifting of a substrate surface state to a different energy (Shwalb et al Phys Rev Lett 101 146801 (2008)). These references are good context that should probably be added to the paper near lines 79-80 where similar issues are addressed about surface state confinement. Moreover, they show the importance of carefully understanding substrate contributions in the present experimental study. The DFT suggest that the substrate does not need to be involved, but I find this argument weak without direct substrate measurements.

Finally, a comment is made in the Summary in lines 311-312 to the effect that there is insufficient evidence that pi-pi interactions "actually happen" in the organic electronics literature. I think this statement is not correct and should be removed. Alternately, a more careful scholarly critique needs to be made if the authors insist on the point. Indeed it may be true that NFE bands from pi-pi interactions are not known but impacts of general pi-pi interactions are VERY well known, leading to measurable bandwidths in ARPES data for rubrene (see e.g. an APL from Yongli Gao's group). This is only a minor recent example from a huge field with a long and complex history.

In summary, on the merits of the stunning experimental data, I think this paper should be published in Nature Communications. It needs a few minor revisions to provide controls for known substrate effects on NFE bands and to remove an inaccurate commentary.

Reviewer #3 (Remarks to the Author):

In this article, Cui et al. report an experimental physical phenomenon that NFE like conduction band can be realized through mediating intermolecular vdW interactions between C60 molecules and BP substrate, which is investigated by STM technique and DFT calculation. The obtained results presented in the manuscript is interesting, however, the innovation and the quality of the work may not fulfil the high standard of Nature Communication.

1. The novelty of this study: The use of such a material (C60/BP) to enhance charge delocalization is quite a common method in this field. The innovation in this work should be further addressed.
2. The role of the similar fullerene: The authors highlighted the role of C60 in the physical phenomenon. Similar fullerene molecules, such as C70 and C78, could be added to for comparison.
3. The stability of the materials: The simple preparation by vdW interactions resulted in the formation of the materials (C60/BP). The stability of the material needs to be discussed.
4. The details of the experiments: The author probed the effect of vdW interactions between C60 molecules and BP substrate on the charge transport properties. The different interactions, such as covalent bond, should be discussed in this material. Some further control experiments may be required.

In addition, there are others minor points:

5. Some of the sentences in the first paragraph of the introduction are ambiguous. For example: "... but is rarely optimal when the noncovalent intermolecular van der Waals forces define the self-assembly and consequently, the intermolecular electronic coupling"
"Because C60 molecules form solid where a balance of repulsive electrostatic and Pauli type, and

attractive London dispersion forces do not enhance intermolecular electronic hybridization..."

"...but only marginal improvement were achieved as the dispersions of the electronic bands were not effectively modified and thus the primary hopping mechanism was not altered."

In the abstract, "hundreds of $\text{cm}^2/\text{V}\cdot\text{s}$ " is inappropriate and the numerical value should be specified.

6. Line 44, "0-3D" should be defined.

7. Line 58, authors should pay attention to font format.

8. Line 73, I think the word "dominant" is more suitable than "define".

9. Line 111, the accuracy of standard deviation should not be less than the standard value.

10. Line 128 and 139, the input symbol format seems incorrect.

11. Line 165, what does (34) stand for?

12. Line 172 and 173, authors cite the same literature continuously, and I recommend merging it.

13. In the method section, line 327, I think the " 600° " is a mistake?

14. The title of the journal is "Nature communications".

I would suggest a careful editing of the paper and submit it to a more specific journal.

Reviewer #1

Reply to Reviewer #1:

The authors sincerely appreciate the Reviewer's nice evaluation for thinking that the experimental finding "is an unconventional feature for molecular semiconductors" and being in favor of this work's publication in Nature Communications. We also greatly appreciate the Reviewer's suggestive comments helping us to improve the manuscript. We reply to these comments in detail.

Comments:

1. In tight binding calculations, the transfer integral (t) between nearest neighbor plays important roles in determining the bandwidth and effective mass. To first order approximation, it should scale with the splitting between the single molecule orbital and collective band from interaction (named NFE in the paper), which is $\sim 0.2\text{eV}$ from STS. What is the calculated value of t ? The effective mass of $0.53m$ is remarkably small, compared to many inorganic semiconductors like MoS₂ ($0.45m$). However, in those inorganic semiconductors, the transfer is one order of magnitude larger than C₆₀ here. Therefore the low effective mass is not well justified. More in-depth discussion about transfer integral, band dispersion and effective mass is needed.

2. Again regarding t , the free standing C₆₀ crystals have inter-molecular distance of 10.04 \AA , while the in current work it is reduced to $10.00 \pm 0.2 \text{ \AA}$. What's the effect of this 4% on t ?

3. What is LUMO a/b/c, are these the lowest LUMOs or just three random orbitals selected by the authors? Why select these three orbitals, and which orbitals do the two prominent peaks near 1.1eV correspond to?

Reply:

As all these three comments are closely related, we answer them in a systematic way in the following.

1. About LUMO-a, -b, -c:

First of all, we regret that we did not make the definition very clear in the original manuscript (MS). The LUMO orbital of an isolated C₆₀ is a triply-degenerate (ignoring the electron spin). LUMO-a, -b, -c derive from the LUMO of a free molecule when the degeneracy is broken by nonisotropic environment such as caused by the intermolecular and molecule-substrate interactions. The LUMO-a, -b, -c label is normally used to identify the three orbitals that can be identified when the perturbation is sufficiently strong, to distinguish their origin from lowest energy state to the higher ones. In Fig. S4 in the Supplementary Information (SI), we show the energy levels of an isolated C₆₀ from DFT calculations. The calculations show that the LUMO state is composed of three degenerate orbitals whose energies are within 3 meV difference, which is within our calculation accuracy. In Fig. 3b, the band structure of a C₆₀ monolayer shows the six LUMO bands that originate from the three degenerate LUMO states due to their different hybridizations within the 2D monolayer. We have made revisions in the MS to make this clearer. The details are shown in the #13, #20 in Revision List for MS and #2 in Revision List for SI.

2. About identifying the C_{60} orbitals with the features in the dI/dV spectra:

We are sorry that in the original MS this point is only briefly discussed in the figure caption of Fig. 3. This might be the reason why it easily causes confusions.

In the MS, Fig. 3b shows the calculated band structure of the C_{60} monolayer with the structures shown in Fig. 1f. The calculation results show the six LUMO bands that originate from the three degenerate LUMO orbitals (as mentioned above) due to their different hybridizations within the 2D monolayer. The lowest unoccupied energy at the Γ point belongs to the LUMO-a bonding band, which results into the shoulder in the DOS plot (marked with the violent circle in Fig. 3a) and corresponds to the shoulder (~ 0.78 eV) recorded in our dI/dV STS spectra (Fig. 2b). Those two higher-energy peaks marked with cyan and light orange circles in the DOS plot shown in Fig. 3a are ascribed to the LUMO-b, -c bonding bands and the LUMO-a, -b, -c antibonding bands, respectively, which reproduce the two major dI/dV peaks at 1.05 eV and 1.33 eV (Fig. 2b). We have revised both the text and the figures to make this point clear. Please see details in #13, #24 in Revision List for MS.

3. About the energy splitting from the single molecule orbital to the formation of the collective band:

Given the two points mentioned above, it could be seen that the energy splitting for the LUMO-a orbital upon forming the band is ~ 0.55 eV, *i.e.*, splitting between ~ 0.78 eV (LUMO-a bonding state) and ~ 1.33 eV (LUMO-a anti-bonding state), according to our dI/dV spectra (Fig. 2b in MS). This is very close to our DFT result of 0.50 eV (Fig. 3b in MS).

According to the Reviewer's suggestion, we performed tight binding calculations to check the energy splitting with transfer integral (t). In our calculations, we obtain the transfer integrals between C_{60} within the framework of the Marcus-Hush two-state model similar to that in the literature (Deng, W.-Q., Goddard, W. A., Predictions of hole mobilities in oligoacene organic semiconductors from quantum mechanical calculations. *J. Phys. Chem. B* **2004**, 108 (25), 8614-8621).

With the two-state model, the transfer integral for a given electronic level is related to the energetic splitting of that level in the dimer as compared to the isolated neutral molecule. Thus, we built a dimer model which has the same inter-molecular distance and orientation as those in the C_{60} film (Fig. 1f). For LUMO-a, it splits into a bonding and an anti-bonding states, so the transfer integral $t \sim 80$ meV is thus obtained from

$$t = \frac{1}{2}(E_{LUMOa-antibonding} - E_{LUMOa-bonding}).$$

In the 2D tight binding model based on the experimentally observed lattice constants (Fig. 1f in MS), the k -dependent energy of an orbital is $E(k_x, k_y) = 4t \cos \frac{k_x a}{2} \cos \frac{\sqrt{3} k_y a}{2}$ (details are provided in Fig. S6 in SI), where t is the transfer integral, a is the lattice constant. The bandwidth of LUMO-a should be $\propto 8t$ and a t of 80 meV would give a 0.64 eV bandwidth. This is basically consistent with the DFT result of 0.50 eV. The model calculated effective masses m^* of 0.96 m_e along the Γ -X (zig-zag) and 0.32 m_e along Γ -Y (arm-chair) directions, using $t=80$ meV, are also consistent with DFT results

of 0.70 m_e and 0.53 m_e , respectively (Fig S6 in SI).

We agree that typically t in the inorganic semiconductor such as MoS₂ is one order of magnitude larger than that for C₆₀. However, when we expand the one dimensional (1D) tight-binding energy expression $E \propto -2t \cos ka$ to the quadratic order,

$$E(k) \propto -2t \cos ka \propto -2t(1 - k^2 a^2), \quad (1)$$

from which we get an expression for the effective mass

$$m^* \propto \frac{1}{\partial^2 E / \partial k^2} \propto \frac{1}{2ta^2} \quad (2)$$

with a the lattice constant.

From Eq. (2), the effective mass is inversely proportional to the product of transfer integral t and square of the lattice constant a . The primitive unit cell of C₆₀ monolayer is larger than that of MoS₂. In the armchair direction, the lattice constant of the primitive unit cell of C₆₀ is 17.2 Å; in the zig-zag direction, the lattice constant of the primitive unit cell of C₆₀ is 10.0 Å; both of which are much larger than that of MoS₂ of 3.3 Å. From this point of view, it is reasonable that the smaller t of C₆₀ can confer an effective mass comparable to that of MoS₂. We have added the discussions about transfer integral, band dispersion and effective mass in the revised MS as shown in the **#14 in Revision List for MS and #3 in Revision List for SI**. We thank the referee for this suggestion to make the MS complete and clear.

4. About the effect of 4% intermolecular distance change on the transfer integral t :

When we increase the inter-molecular distance by 4%, the transfer integrals t decreases to 51 meV. We point out that the lattice constant changes by 4%, but the change in the intermolecular C-C distance is 3-4 times larger in terms of percent. Considering that the wave function overlap increases approximately exponentially with the distance, the layer compression can have a pronounced effect. We also need to mention that in addition to the inter-molecular distance, the mutual orientation angle (*h:h-h:h* angle as shown in Fig. 1d in MS) between the two C₆₀ also plays an important role in determining t . When the *h:h-h:h* angle is 0°, transfer integral t further reduces to 29 meV. All these results indicate that both the intermolecular distance and the relative orientations are crucial for the formation of the NFE LUMO-a band found in this work, which are consistent with our DFT results presented in Fig. 4b, 4c and 4d in MS.

Comments:

4. I notice that the “arms” in experiments have different angles from theoretical calculations (see the difference between Fig. 2d inset and Fig. 3c), what is the origin of such inconsistency?

Reply:

We greatly appreciate that the reviewer noticed this issue, which was not well-described in the original MS. The inconsistency comes from the improper positions of the “dots” that are utilized as marks to guide eyes. Please see the revised Fig. 2e inset and Fig. 3c in MS where the locations of the “dots” are re-arranged. It could be seen that the experimental observations and theoretical results are quite consistent. Please see details in **#21, #23 in Revision List for MS**.

Comments:

5. In calculating the delocalization of orbitals the substrate BP was not included, and therefore the substrate-molecule interaction was neglected. However, according to the calculations this interaction is much stronger than inter-molecular interactions, especially for LUMO-b and c according to fig. 3e. The authors claim that “The band structure calculations of the C₆₀ monolayer are performed without the BP substrate because they are nearly identical in the band gap region”, however Fig. S2 does not show band structure calculations with BP substrate.

Reply:

We thank the reviewer for reminding us this important point. The calculations were performed for both the C₆₀ monolayer with BP substrate and without BP substrate by utilizing the supercell shown in Fig. 1f. In MS, we provide the DFT results of the isolated C₆₀ monolayer without BP substrate to clearly show the dispersion of the C₆₀ LUMO bands. The DFT results of C₆₀ monolayer with BP substrate are provided in SI. Please see the following detailed explanations.

(I) The differential charge density analysis for C₆₀ monolayer/BP shows that there is negligible charge transfer between BP and C₆₀ (Line 134th-138th in MS). This is consistent with the STS spectra which show that C₆₀ on the BP surface is still semiconducting with a ~2.8eV gap (Fig. 2b inset in MS). This leads us to conclude that the interactions between C₆₀ and BP mainly involve vdW forces.

(II) We have compared the DFT calculated DOS of C₆₀ monolayer with and without the BP substrate by plotting the partial density of state (PDOS) of the BP substrate and the C₆₀ monolayer (Fig. S3 in SI). The PDOS of the C₆₀ monolayer is similar with that of the isolated C₆₀ lattice (Fig. 3a in MS). These results also show that there is negligible hybridization between the C₆₀ and BP substrate.

(III) Furthermore, when performing dI/dV spectra from the edge of the C₆₀ monolayer island to the BP substrate, the transition of the spectra shows the gradual disappearance of the C₆₀ feature and the appearance the BP conduction band (the supplemented Fig. 2d in MS as Reviewer 2 suggested). There is no indication of appreciable electronic hybridization between BP and C₆₀. Based on these results, we conclude that there are negligible electronic interactions between BP and C₆₀.

(IV) As we demonstrate in Fig. S2 in SI, removal of the substrate does not appreciably change the energies and dispersions of the six LUMO bands. We plotted the band structure of C₆₀ adsorbed on the BP substrate and marked the bands with significant projections on C₆₀ molecules in hot colors while those purely BP bands are not shown (Fig. S2a in SI). By comparing the band structure of the LUMO and HOMO bands of C₆₀ assembles with and without (Fig. S2b in SI) the BP substrate, we found there is no obvious difference between them. However, the energy level of the LUMO-a bands at the Γ point in the case with BP substrate crosses the E_f in the band structure. This is inconsistent with the dI/dV measurement and is, most likely caused by the underestimated band gaps using DFT.

With all the above results, we conclude that the BP substrate does not appreciably disturb the electronic structures that we are interested in, namely the NFE band around the band gap of C₆₀ assembles. Therefore, DFT results of the isolated C₆₀

monolayer without substrate essentially capture the main physics of the observed phenomena. As we showed in Fig. S2a in SI, inclusion of the BP substrate in the electronic structure calculation may introduce several substrate bands in the proximity of the LUMO bands, which distract from the main physics we intended to present for the NFE LUMO-a band. For more clearly showing the dispersions of the LUMO bands, the band structure was plotted with the fully relaxed C₆₀ monolayer without BP substrate. We are sorry we did not clarify this point explicitly in the original MS. We have modified the text and also provide the band structure of the C₆₀ assemblies with the BP substrate in the revised SI. Please see **#12, #22 in Revision List for MS and #1 in Revision List for SI** for details. We thank the referee quite a lot for carefully reviewing our work and helping us to improve the MS.

Comments

6.Minor point: page 6 last line, green should have been blue.

Reply: We have corrected it in the revised MS. Please see **#11 in Revision List for MS** for details.

Reviewer #2*Reply to Reviewer #2:*

The authors are really honored with the Reviewer 2's high praise for this work by describing the experimental findings as "remarkable" and "stunning". We also highly appreciate that Reviewer 2 recommends that the work "should be published in Nature Communications". According to the two major concerns from Reviewer 2, we list our responses in detail in the following.

Comments:

1. "Prior to publication, I recommend that the authors include STS spectra measured on the black phosphorous substrate. It is a glaring omission to not see this data, particularly in Figure 2c where the position-dependent spectroscopy stops at the edge of the C₆₀ layer. It should be extended to the BP region beyond for direct comparison."

Reply:

We thank the reviewer for this recommendation. We performed complementary experiments as the reviewer suggested and added the data to Fig. 2 in MS.

In the revised MS, we supplemented a new figure showing the position-dependent spectra by extending the measurement locations from the C₆₀ island to the BP region (Fig. 2d in MS). It can be seen that the transition of the spectra shows the gradual disappearance of the C₆₀ features and the appearance the BP states, which are distinct and particular to each region. By combing with other results (please see details of the reply to comment 5 to Reviewer #1): 1) C₆₀ molecules within the monolayers have semiconductor band structure with a ~2.8 eV gap (Fig. 2b inset in MS); 2) The DFT calculated differential charge analysis at the C₆₀-BP interfaces for C₆₀/BP does not show charge transfer between C₆₀ and BP Line 134th-138th in MS; 3) The PDOS of C₆₀ for C₆₀/BP does not show electronic contribution from BP (Fig. S3 in SI); 4) The calculated band structures of the LUMO and HOMO states of C₆₀ assemblies with and without BP substrate are essentially the same (Fig. S2 in SI), we conclude that the substrate is electronically inert to those states dominating transport properties of the C₆₀ monolayers. Please see #10, #22 in Revision List for MS for details.

Comments:

2. "On a related note, earlier reports of NFE behavior in molecular monolayers (Temirov et al Nature 444, 350 (2006)) were erroneously attributed to molecules alone and were only later attributed to a shifting of a substrate surface state to a different energy (Shwalb et al Phys Rev Lett 101 146801 (2008)). These references are good context that should probably be added to the paper near lines 79-80 where similar issues are addressed about surface state confinement. Moreover, they show the importance of carefully understanding substrate contributions in the present experimental study. The DFT suggest that the substrate does not need to be involved, but I find this argument weak without direct substrate measurements".

Reply:

We thank the reviewer for this recommendation. Because the NFE band discovered

by Temirov originates from a surface state of the metallic substrate, as Shwalb pointed out later, this is totally different from our case where the NFE band comes from the adsorbed molecular overlayer itself. In the original MS, we have discussed this briefly. As Temirov's work is the first one showing NFE bands in organic monolayers, even though this property was provided by the metal substrate, the reviewer is quite right that original references should be added, even though the current understanding of the NFE state found by Temirov is explained by Tautz in Ref. 1 in MS. In the revised manuscript, we 1) supplement the related references (Temirov *et al.*, *Nature* **444**, 350 (2006); Shwalb *et al.*, *Phys Rev Lett.* **101**, 146801 (2008); Ji *et al.*, *Phys. Rev. B* **77**, 113406 (2008)); 2) provide the complete experimental data in the revised MS (Fig. 2d in MS) as Reviewer 2 suggested, to show that the electronic state of BP does not contribute to the LUMO-a NFE band. We hope that these revisions clearly define the difference between Temirov's original report and our findings. Please see **#8 and #19 in Revision List for MS** for details.

Comments:

3. *About the inaccurate commentary in the summary: "Finally, a comment is made in the Summary in lines 311-312 to the effect that there is insufficient evidence that pi-pi interactions "actually happen" in the organic electronics literature. I think this statement is not correct and should be removed. Alternately, a more careful scholarly critique needs to be made if the authors insist on the point. Indeed it may be true that NFE bands from pi-pi interactions are not known but impacts of general pi-pi interactions are VERY well known, leading to measurable bandwidths in ARPES data for rubrene (see e.g. an APL from Yongli Gao's group). This is only a minor recent example from a huge field with a long and complex history".*

Reply:

We thank the reviewer for this comment and have removed the original statement. The new description emphasizes that our work reveals that π - π interactions can produce the NFE band, which is not expected before. Please see **#15 in Revision List for MS** for details.

Reviewer #3*Reply to Reviewer #3:*

The authors thank Reviewer #3 for the comments that “obtained results presented in the manuscript is interesting”. Also, we really appreciate that the Reviewer read the manuscript very carefully, and offered suggestive comments, as well as pointing out spelling typos. However, we feel very sorry that the original MS did not convince the Reviewer about the novelty of this work. We hope the revised MS, which significantly improved based on the Reviewers’ comments, can help to convince the Reviewer. Please see the detailed responses as follows.

Comments:

1. The novelty of this study: The use of such a material (C60/BP) to enhance charge delocalization is quite a common method in this field. The innovation in this work should be further addressed.

Reply:

We agree with the reviewer that utilizing the substrate to modify the arrangement and electronic structures of the adsorbates is a very common method. However, by utilizing such a common method, the experimental findings discovered in this work, as Reviewers #1 and #2 have pointed out, are “unconventional” and “stunning”. Furthermore, there has been no study using semiconducting BP template to modify intermolecular interactions. We supplement the following statements for explaining the novelty.

(I) First, the NFE LUMO band is unexpected in a typical molecular semiconductor, and has not been found before. As Reviewer #2 mentioned, “*On a related note, earlier reports of NFE behavior in molecular monolayers (Temirov et al Nature 444, 350 (2006)) were erroneously attributed to molecules alone and were only later attributed to a shifting of a substrate surface state to a different energy (Shwalb et al Phys Rev Lett 101 146801 (2008))*”, NFE behaviors discovered in previous work originate from the metallic substrate, rather than from the molecules. Our work, to the best of our knowledge, is the first one showing that the NFE LUMO can be realized at the conduction band minimum of a molecular semiconductor through direct interactions between conventional molecular orbitals [Our previous work on the NFE band formed by correlation bound superatom molecular orbitals (Feng M. *et al.*, *Science* **320**, 359 (2008)) is far above the LUMO bands].

(II) Second, revealing that the NFE LUMO band originates from the intermolecular π - π is particularly surprising. Even though π - π interactions are very well known to impact the dispersion of a band structure, it is the first time to show that π - π interactions could be realized to produce a NFE molecular band. This NFE band confers calculated carrier mobility of ~ 200 to 440 $\text{cm}^2/\text{V}\cdot\text{s}$ in C_{60} monolayers, a value which is of an order higher than the maximum mobility realized in this materials and is orders higher than in other commonly studied organic semiconductors. The fact that the π - π interactions could enable the NFE like electronic band formation provides a new strategy for tailoring electronic properties in organic materials.

(III) Another important point is that our work shows that fullerene molecules, even though ~34 years after their discovery, are still surprising finding that justify their early promise. Our work shows that if C₆₀ molecules are arranged by the substrate, their transport properties can be tailored in a totally different way from previous methods (through the weak vdW interactions rather than other strong interactions like charge-transfer). This finding provides a new strategy for tailoring remarkable electronic properties, such as the wealthy electronic phase transitions, of this novel material.

The novelty of this work is expressed in Line 27th-32th; 34th-37th; 82th-84th; 165th-167th; 263th-264th; 326th-328th; 343th-346th; 346th-347th; 348th-350th; 351th-355th. Hope the above descriptions could help clarifying the Reviewer's concerns.

Comments:

2. *The role of the similar fullerene: The authors highlighted the role of C₆₀ in the physical phenomenon. Similar fullerene molecules, such as C₇₀ and C₇₈, could be added to for comparison.*

Reply:

We thank the Reviewer for this suggestion. In this MS, we show that a delicate balance between C₆₀-C₆₀ and C₆₀-BP, through modifying the intermolecular distance and orientations, could produce the NFE bands through intermolecular interactions. BP might not be a good substrate for C₇₀ and C₇₈ because their different sizes and symmetries would produce entirely different interactions. However, we believe similar phenomenon would take place for these fullerenes if proper substrates are chosen.

Comments:

3. *The stability of the materials: The simple preparation by vdW interactions resulted in the formation of the materials (C₆₀/BP). The stability of the material needs to be discussed.*

Reply:

We are grateful to the reviewer for this suggestion. The reviewer is right that the sample is prepared using a simple method. C₆₀ molecules are deposited onto the BP surface at room temperature through a vapor evaporation method. Then the sample is cooled down to liquid nitrogen (LN₂) temperature for the measurements. The adsorption structure of C₆₀ molecules within the monolayers is very stable at LN₂ temperature. Structure phase transitions are known to occur in C₆₀ solids: at 260 K, a first-order phase transition occurs from the face-centered-cubic crystal to a simple cubic structure; at 90 K, the molecules rearrange to attain the best global minimum and is designated as a pseudo-face-centered-cubic structure, with the rotational degrees of freedom being frozen. In our case, for C₆₀ on BP, as the vdW interactions between BP and C₆₀ are stronger than that for C₆₀-C₆₀, and we therefore expect that the adsorption structure of the molecules is frozen at temperatures above 90 K. In the case of the stability of the materials themselves, as BP is synthesized at a temperature higher than 700 K and C₆₀ molecules are evaporated at a temperature about ~650 K, the materials are very stable in this point of view. We have added the associated discussion in the revised MS and please see **#17 in Revision List for MS** for details.

Comments:

4. *The details of the experiments: The author probed the effect of vdW interactions between C₆₀ molecules and BP substrate on the charge transport properties. The different interactions, such as covalent bond, should be discussed in this material. Some further control experiments may be required.*

Reply:

We thank the Reviewer for this valuable suggestion. Covalent bond could be formed between C₆₀ molecules within the solids when under external perturbations as high pressure, UV light irradiation, doping or excitation of tunneling electrons. The carrier mobility of C₆₀ polymer, however, is very low, e.g. 0.068 cm²/V·s (*Phys. Rev. B* **75**, 075203 (2007)). These are known results and we add the related references in the introduction. Please see **#4, #5 and #18 in Revision List for MS** for details. Our experiments and theory show no evidence of covalent bond formation, and the NFE properties do not require it.

Comments:

5. *Some of the sentences in the first paragraph of the introduction are ambiguous. For example:*

“... but is rarely optimal when the noncovalent intermolecular van der Waals forces define the self-assembly and consequently, the intermolecular electronic coupling”

“Because C₆₀ molecules form solid where a balance of repulsive electrostatic and Pauli type, and attractive London dispersion forces do not enhance intermolecular electronic hybridization...”

“...but only marginal improvement were achieved as the dispersions of the electronic bands were not effectively modified and thus the primary hopping mechanism was not altered.”

In the abstract, “hundreds of cm²/V·s” is inappropriate and the numerical value should be specified.

Reply:

“... but is rarely optimal when the noncovalent intermolecular van der Waals forces define the self-assembly and consequently, the intermolecular electronic coupling”

“Because C₆₀ molecules form solid where a balance of repulsive electrostatic and Pauli type, and attractive London dispersion forces do not enhance intermolecular electronic hybridization...”

With these sentences, we try to express the idea that in a vdW solid, the vdW forces determine the arrangement of the molecules, and this may or may not be related with an optimal intermolecular electronic coupling which is the main factor determining the charge transport. We modified the sentences as the following trying to make them clear. *“As a typical van der Waals material, C₆₀ molecules form solid through a balance of the Pauli type, electrostatic repulsive forces and the attractive van der Waals forces, which, however, do not enhance intermolecular electronic hybridizations...”* Please see **#3 in Revision List for MS** for details.

“...but only marginal improvement were achieved as the dispersions of the electronic bands were not effectively modified and thus the primary hopping mechanism was not altered.”

This sentence, combined with the previous sentences, try to express the idea that 1) only when dispersive electronic bands could be formed within fullerene materials, the way of charge transport within the materials could be changed from the primary hopping mechanism to the band transport mechanism. This will be a big advantage for the charge transport; 2) in the efforts of functionalization of fullerenes by synthesizing fullerene derivatives only limited progress was achieved as no dispersive electronic bands were realized. We modified the sentences as below trying to avoid the confusion here. *“Functionalization of C₆₀ by synthesizing fullerene derivatives has been investigated as a means to improve its electron transport, but the improvement was limited as the dispersions of the electronic bands were not effectively modified.”*¹¹⁻¹⁴ Please see **#4, #5, #6 and #18 in Revision List for MS** for details.

In the abstract, “hundreds of cm²/V·s” is inappropriate and the numerical value should be specified.

The numerical value has been specified. Please see **#1 in Revision List for MS** for details.

Comments:

6. Line 44, “0-3D” should be defined.
7. Line 58, authors should pay attention to font format.
8. Line 73, I think the word “dominant” is more suitable than “define”.
9. Line 111, the accuracy of standard deviation should not be less than the standard value.
10. Line 128 and 139, the input symbol format seems incorrect.
11. Line 165, what does (34) stand for?
12. Line 172 and 173, authors cite the same literature continuously, and I recommend merging it.
13. In the method section, line 327, I think the “600°” is a mistake?
14. The title of the journal is “Nature communications”.

Reply;

We have corrected these typos and mistakes in the revised manuscript as shown in the **#2, #7, #9 and #16 in Revision List for MS**.

Line 44, “0-3D” should be defined.

We have defined “0-3D” in the revised MS.

Line 58, authors should pay attention to font format.

Line 128 and 139, the input symbol format seems incorrect.

Thanks. The problems might be caused by conversion between different software. The format problems have been corrected.

Line 165, what does (34) stand for?

Line 172 and 173, authors cite the same literature continuously, and I recommend merging it.

Thanks. The “(34)” is the reference number. We have corrected this. We also merged the reference the reviewer suggested.

REVISION LIST:

Revisions List for Manuscript (MS):

- #1. ----- Line 36th: “~200 to 440 cm²/V·s”
- #2. ----- Line 47th: “zero- to three-dimensional (0-3D)”
- #3. ----- Line 48th-Line 50th: “As a typical vdW material, C₆₀ molecules form solids through a balance of the Pauli type, electrostatic repulsive force and the attractive vdW force,¹⁰ which, however, does not enhance intermolecular electronic hybridizations.”
- #4. ----- Line 52th-54th: “Functionalization of C₆₀ by synthesizing fullerene derivatives has been investigated as a means to improve its electron transport, but the improvement was limited as the dispersions of the electronic bands were not effectively modified.¹¹⁻¹⁴”
- #5. ----- Line 55th: *Ref. 14* is added (the order of the references in the following changed accordingly).
- #6. ----- Line 56th: “increase the intermolecular electronic hybridizations.”
- #7. ----- Line 77th: “dominate”
- #8. ----- Line 82th-84th: “NFE dispersion has been attributed to π orbitals of a molecular monolayer on metallic substrates, but its origin has later been reassigned to quantum confinement of a metal surface state, rather than the electronic hybridization among organic molecules.²⁴⁻²⁸”, and *new references 26, 27 and 28* are added.
- #9. ----- Line 117th and 120th: “ $9.9 \pm 0.2 \text{ \AA}$ ”, “ $10.0 \pm 0.2 \text{ \AA}$ ”
- #10. ----- Line 187th-190th: “STS spectra are also acquired on the locations across the C₆₀/BP island edge, as shown in Fig. 2d. It is evident that the spectroscopic shoulder feature of C₆₀ within the island terminates at the edge C₆₀ molecules; after leaving the C₆₀ island edge, the spectra show the typical bare BP character.²¹”
- #11. ----- Line 199th: “blue”
- #12. ----- Line 209th-221th: “The calculations were performed for both the C₆₀ monolayers with BP substrate and without BP substrate using the supercell shown in Fig. 1f. As shown in Fig. S2 in SI, the energies and dispersions of the LUMO bands of two C₆₀ molecules in the supercell does not appreciably change upon the removal of BP substrate. Furthermore, removing the BP substrate does not cause appreciable change in the partial density of states of C₆₀ monolayer (Fig. S3 in SI). In light of these comparisons, we conclude that DFT results of the isolated C₆₀ monolayer without BP substrate fully capture the essential features of the NFE band formation. As shown in Fig. S2a in SI, the inclusion of the BP substrate in the electronic structure calculation introduces several bands in the proximity of the C₆₀ LUMO bands, which may blur the NFE LUMO-a band we intended to present. Therefore, we only provide DFT results of the fully relaxed C₆₀ monolayer without BP substrate in the following discussions.”
- #13. ----- Line 226th-243th: “For an isolated C₆₀, the LUMO state is triply-degenerate (Fig. S4 in SI).³⁹ LUMO-a, -b, -c derive from the LUMO of a free molecule when the degeneracy is broken by nonisotropic environment such as caused by the intermolecular and molecule-substrate interactions. The LUMO-a, -b, -c label is normally used to identify the three orbitals that can be identified when the perturbations are sufficiently strong, to distinguish their origin from lowest energy state to the higher ones.⁴⁰ Here, the supercell contains two C₆₀ molecules and those

LUMO orbitals thus form six LUMO bands (three pairs of bonding - antibonding bands) in the monolayer. The dispersion and the energy levels of those bands are determined by C_{60} - C_{60} interactions within the monolayer. Figure 3b shows the calculated band structure of C_{60} monolayer with the structures shown in Fig. 1f. The lowest unoccupied energy at the Γ point belongs to the LUMO-a bonding band, which results into the shoulder in the DOS plot (marked with the violet circle in Fig. 3a) and corresponds to the shoulder (~ 0.78 eV) recorded in our dI/dV STS spectra (Fig. 2b). Those two higher-energy peaks marked with cyan and light orange circles in the DOS plot shown in Fig. 3a are ascribed to the LUMO-b, -c bonding bands and the LUMO-a, -b, -c antibonding bands, respectively, which reproduce the two major dI/dV peaks at 1.05 eV and 1.33 eV (Fig. 2b).”, and two new references (Refs. 39, 40) are added.

#14. ----- Line 248th-251th: “We further confirm the rather small DFT calculated m^* in C_{60} monolayer by performing complementary tight binding calculations (Fig. S6 in SI). These calculations give a bandwidth of 0.64 eV for the LUMO-a band and m^* values of 0.96 m_e along the Γ -X (zig-zag) and 0.32 m_e along Γ -Y (arm-chair) directions.”

#15. ----- Line 343th-346th: “In the literature on organic electronics, there are many discussions of π - π interactions, and how they might enhance charge transport, without much actual evidence that NFE band could be induced by π - π interactions.”

#16. ----- Line 362th: “600°C”

#17. ----- Line 368th-369th: “The adsorption structure of C_{60} molecules within the monolayers is stable at LN₂ temperature.”

#18. ----- Line 471th-473th: “14. Dzwilewski, A., Wagberg, T. & Edman, L. C_{60} field-effect transistors: Effect of polymerization on electronic properties and device performance. *Phys. Rev. B* **75**, 075203 (2007).”

#19. ----- Line 502th-507th: “26. Temirov, R., Soubatch, H., Lucian, A. & Tautz, F. S. Free-electron-like dispersion in an organic monolayer film on a metal substrate. *Nature* **444**, 350 (2006). 27. Schwalb, C. H. *et al.* Electron lifetime in a shockley-type-metal-organic interface state. *Phys. Rev. Lett.* **101**, 146801 (2008); 28. Ji, W., Lu, Z. Y. & Gao, H. J. Multichannel interaction mechanism in a molecule-metal interface. *Phys. Rev. B* **77**, 113406 (2008).”

#20. ----- Line 528th-532th: “39. Haddon, R. C. Electronic structure, conductivity and superconductivity of alkali doped C_{60} . *Acc. Chem. Res.* **25**, 127-133 (1992); 40. Silien, C., Pradhan, N. A., Ho, W. & Thiry, P. A. Influence of adsorbate-substrate interaction on the local electronic structure of C_{60} studied by low-temperature STM. *Phys. Rev. B* **69**, 115434 (2004).”

#21. ----- Line 673th: “Figure 2d is supplemented” and “the inset in Fig. 2e” is modified by re-arranging the locations of the dots marks.

#22. ----- Line 697th-700th: “(d) Position-dependent STS spectra acquired on the locations across the C_{60} /BP edge. The light-green rectangle highlights the energy region where the spectroscopic shoulder feature is observed.”

#23. ----- Line 713th: “Fig. 3a and Fig. 3b” are modified by adding marks; “Fig. 3c” is modified by re-arranging the locations of the dots marks.

Revisions List for Supplementary Information (SI):

- #1.** ----- Line 58th: A new Supplementary Section “Fig. S2 and related descriptions” are supplemented to further describe “DFT calculated band structure of C₆₀/BP and isolate C₆₀ monolayer”.
- #2.** ----- Line 107th: A new Supplementary Section “Fig. S4 and related descriptions” are supplemented to further describe “The energy levels of the LUMO state of an isolated C₆₀ from DFT calculations”.
- #3.** ----- Line 171th-224th: A new Supplementary Section “Fig. S6 and related descriptions” are supplemented to further describe “Tight binding calculations for the C₆₀ monolayer with the experimentally observed lattice constants”.
- #4.** ----- Line 325th: A new Supplementary reference is supplemented. “1. Deng, W. - Q., Goddard, W. A., Predictions of hole mobilities in oligoacene organic semiconductors from quantum mechanical calculations. *J. Phys. Chem. B* **108**, 8614-8621, (2004)”.

REVIEWERS' COMMENTS:

Reviewer #1 (Remarks to the Author):

The authors have addressed my comments. I recommend publication in Nature Comm. in the present form.

Reviewer #2 (Remarks to the Author):

The authors have addressed my comments with edits to the manuscript and have added new control measurements that help clarify their ideas.

I recommend publication without further revision.

Reviewer #3 (Remarks to the Author):

The authors have addressed my concerns, and I recommend this manuscript to publish in Nature Communications.